# Guideline Adherence and Outcomes of Patients with Candidemia in Brazil

**DOI:** 10.3390/jof10040282

**Published:** 2024-04-12

**Authors:** Jordana Machado Araujo, João Nóbrega de Almeida Junior, Marcello Mihailenko Chaves Magri, Silvia Figueiredo Costa, Thaís Guimarães

**Affiliations:** 1Infection Control Department, Hospital das Clínicas, University of São Paulo, São Paulo 05403-900, Brazil; m.jordana01@gmail.com; 2Central Laboratory Division, Hospital das Clínicas, University of São Paulo, São Paulo 05403-900, Brazil; jnaj99@gmail.com; 3Infectious Diseases Department, Hospital das Clínicas, University of São Paulo, São Paulo 05403-900, Brazil; marcello.magri@hc.fm.usp.br (M.M.C.M.); silviacosta@usp.br (S.F.C.)

**Keywords:** candidemia, epidemiology, mortality, guidelines

## Abstract

Candidemia is a significant cause of mortality among hospitalized patients, both worldwide and in Brazil. Prompt and appropriate treatment are essential to mitigate mortality, and clinical practice guidelines aim to optimize patient care based on the best scientific evidence. This study aims to examine the management of candidemia, assessing adherence to the guidelines of the Brazilian Society of Infectious Diseases in a single center located at São Paulo, Brazil. All adult patients hospitalized from 2016 to 2018 who presented one positive blood culture for *Candida* spp. were included. Electronic medical records were retrospectively reviewed to collect information relevant to the treatment for candidemia, in order to assess the adherence to the Brazilian guideline for the management of candidemia in relation to nine defined outcomes, and we correlated those findings with 30-day mortality by using uni- and multivariate analyses. A total of 115 patients were included; 68 patients (59.1%) were male, with a mean age of 55 years. *C. albicans*, *C. tropicalis* and *C. glabrata* were the most prevalent species. In total, 80 patients (69.5%) received antifungal treatment. The adherence to Brazilian guideline recommendations was determined as described in the following: initial treatment with echinocandin in 48 (60%); step-down to fluconazole in 21 (26.2%); collection of first control blood culture in 43 (58.9%); collection of second control blood culture, if the first one had been positive, in 14 (73.6%); treatment for 14 days after the first negative blood culture in 53 (65.4%); central venous catheter (CVC) removal in 66 (82.5%); CVC removal if the first control blood culture had been positive in 17 (89.4%); performance of a transthoracic echocardiogram in 51 (63.7%) and performance of a fundoscopy in 59 (73.7%). Univariate analysis showed that CVC removal and initial echinocandin therapy were more prevalent in the surviving group, but with no statistically significant difference. On the other hand, step-down to fluconazole demonstrated higher survival rate in the multivariate analysis OR 0.15 (95% CI 0.03–0.8); *p* = 0.02. The analysis of these nine recommendations demonstrates that it is necessary to improve adherence to specific recommendations and also disseminate strategies of the initial use of echinocandin as the drug of choice and addressing length of treatment and follow-up and complementary exams. Our study provides reassurance that the step-down to fluconazole is safe and may be recommended, if the preexisting conditions are present.

## 1. Introduction

Candidemia is a significant cause of health care-associated infection, and is associated with high rates of morbidity and mortality among hospitalized patients worldwide [1,2].

Mortality rates are also notably higher in Brazil when compared to those in the Northern Hemisphere, reaching as high as 58.9% in a recent 21-year series of candidemia compared to 20.2% in a multicenter study from Spain [3,4].

Early recognition and prompt initiation of appropriate systemic antifungal therapy are essential weapons to mitigate invasive candidiasis mortality. Clinical practice guidelines aim to provide evidence-based information for the best treatments in order to reduce mortality and synthesize practical recommendations, aiming to optimize patient care in different scenarios.

Several aspects of candidemia management, such as initial echinocandin treatment or early catheter withdrawal, have shown a survival advantage and have been included in the guidelines for candidemia management developed by different societies [5,6,7]. These three guidelines, published in different years, have similar recommendations regarding initial echinocandin therapy, blood culture collection, treatment duration and search for infection acute spread, with the possible step-down to fluconazole differing only with respect to the time to do so.

The Brazilian guidelines for the management of candidiasis published in 2013 recommend multiple best practices for managing patients with candidemia, such as appropriate initial antifungal treatment with echinocandin; repeat blood cultures to confirm bloodstream clearance; treatment duration of at least 14 days from bloodstream clearance; central venous catheter removal, especially if the patient’s blood culture remains positive despite adequate treatment; and ophthalmological and cardiac examination for screening of endophthalmitis and endocarditis [7]. Several of these recommendations are controversial and based on expert opinion, and although there are some practical aspects of the clinical management of candidemia that deserve reconsideration and updated recommendations, adherence to guidelines may improve outcomes and it can be a great opportunity for antifungal stewardship [8].

This study aims to examine the management of candidemia at our institution, assessing adherence to the current guidelines of the Brazilian Society of Infectious Diseases [7].

## 2. Methods

### 2.1. Study Design

A single-center retrospective study was carried out at Instituto Central of the Hospital das Clínicas (ICHC) of University of São Paulo, from 1 January 2016 to 31 December 2018. The study was submitted to the Ethics and Research Committee and approved under number CAAE 19921919.9.0000.0068.

The ICHC is a tertiary teaching hospital with 910 beds, 97 of which are for Intensive Care Units. It also has clinical and surgical inpatient units for various specialties, as well as solid-organ (kidney and liver) and hematopoietic stem-cell transplant services.

### 2.2. Eligible Patients

All patients who presented at least one positive blood culture for *Candida* spp. were included. Only the first episode of candidemia was considered for the clinical characterization of the cases. Relapse was determined if the patient had a negative blood culture during follow-up and later had a positive blood culture with the same species of *Candida* spp.

Outpatients, patients hospitalized in neonatology, patients with blood cultures collected only from a central venous catheter and without paired peripheral blood cultures or with negative peripheral blood cultures, patients with positive blood culture on admission from another institution and patients whose medical records could not be located were excluded.

### 2.3. Data Collection

Electronic medical records were retrospectively reviewed to collect the following information: demographic data; medical history; risk factors for candidemia; microbiological data; therapeutic measures and complementary exams (echocardiogram and fundoscopy); device management and evaluation of the clinical outcome (discharge or death during hospitalization). Mortality was considered to occur within 30 days after diagnosis of candidemia.

### 2.4. Outcomes

We assessed the adherence to the Brazilian guidelines for the management of candidemia, which include the following considerations: appropriate initial antifungal treatment with echinocandin, step-down to fluconazole in sequential therapy to complete a minimum period of 14 days of treatment after determining the etiological agent (species known to be susceptible to azoles) and upon documentation of a favorable clinical and microbiological response to treatment with echinocandins (i.e., clinical stability and documentation of negative blood culture); repeat blood cultures to confirm bloodstream clearance, treatment duration of at least 14 days from bloodstream clearance, central venous catheter removal, and ophthalmological and cardiac examination for screening of endophthalmitis and endocarditis [7].

### 2.5. Statistical Analysis

Quantitative variables were reported as means and categorical variables were reported as absolute numbers and percentages. All information regarding the patients was stored in a database, using Excel 5.0. The analysis of qualitative variables was performed using a chi-squared test or Fisher’s exact test, and ANOVA was used for quantitative variables. The potential factors related to mortality were compared by univariate analysis and all factors identified as significant were submitted to multivariate analysis by the multiple logistic regression model. The independent variables were expressed through their risk ratios and their respective confidence intervals (CIs) of 95% were estimated. All probabilities of significance presented were determined considering a significance level of 0.05 or 5.0%. Statistical calculations for univariate and multivariate analysis were performed using EPI-INFO version 7.2.

## 3. Results

132 patients were initially selected, but 17 were excluded for presenting only catheter blood culture or being without paired peripheral blood culture or with a negative peripheral blood culture, resulting in a total of 115 patients for analysis. No relapse was observed.

Table 1 demonstrate clinical characteristics and risk factors found in patients with candidemia during the study period.

The microbiological analysis of the species showed eight different species of *Candida* spp., with the most common being *Candida albicans* (44; 38.2%), followed by *Candida tropicalis* (24; 20.8%). *Candida glabrata* was responsible for 23 (20%) of the cases and *Candida parapsilosis* for 18 (15.6%). Other species were responsible for six (5.1%) of the cases, with two cases of *Candida krusei*, two cases of *Candida kefyr*, one case of *Candida dubliniensis* and one case of *Candida haemulonii.*

Regarding treatment, 80 patients (69.5%) received antifungal treatment, and 35 (30.4%) were not treated. Only four (3.5%) did not receive any treatment. In 31 patients (26.9%) the reason for non-treatment was death within 3 days of blood culture collection.

The average time to start treatment, from the collection of blood cultures, was 3.5 days, ranging from zero to 19 days, with zero the start of treatment simultaneous with blood culture collection.

Of the patients treated, the antifungal of choice as initial therapy was an echinocandin in forty-eight (60%), fluconazole in thirty-one (38.7%) and voriconazole in one (1.2%) of the cases.

Forty-seven patients (58.7%) maintained the same antifungal agent during the entire treatment period and thirty-three (41.2%) had the regimen changed. Of these, 14 (42.2%) were de-escalated to fluconazole. In relation to the others, ten (30.3%) were guided by culture, six (18.1%) were switched due to the patient’s severity or clinical worsening, two (6%) were due to adverse events, and one (3%) switched to maintain endocarditis treatment.

The mean treatment time was 16 days (median 15 days), ranging from 1 to 42 days. Among the patients who had a change of regimen, it occurred an average of 9 days after the start of treatment.

Of the 80 patients who received treatment, 66 (82.5%) had a catheter removed; the tip was sent for culture in 58 cases (87.8%), with a positive result for *Candida* spp. species in 24 (41.3%). In only two cases were the specie isolated at the tip culture different from the specie isolated in the blood culture.

The time between candidemia and catheter removal ranged from 0 to 21 days (mean 4.3 days), with zero removals simultaneous with blood culture collection. In fifty-one patients (77.2%) the CVC was removed within 72 h of a positive blood culture.

Among the eighty patients who received treatment, fifty-one (63.7%) received transthoracic echocardiography (TTE) as an initial screening for endocarditis, with one (1.8%) having a positive result for infective endocarditis (IE), while in nine patients (11.2%) a transesophageal echocardiogram (TEE) was performed for screening, with one case determined to be positive for IE. Of the patients who underwent a TTE for screening and had no determination of IE, eight patients were latter subjected to transesophageal echocardiography, with two cases of IE being identified, bringing the total to 4/81 (4.9%) cases of IE, with a total of two deaths.

The time to performance of the echocardiogram ranged from −5 to 28 days (that is, some patients received echocardiography before positive blood cultures) by judgment of the attending physician, with a median of 6 days.

Likewise, among the 80 patients undergoing treatment, 59 (73.7%) had an eye fundoscopy, and only one was positive (incidence of endophthalmitis of 1.7%) while showing signs of embolization. This one case had a positive outcome. The time to perform fundoscopy ranged from 2 to 53 days (median of 5 days).

Among the 115 episodes of candidemia, in 68 (58.2%) the outcome was in-hospital mortality. The time between the first positive blood culture and death ranged from zero to 179 days (median of 15 days), with 59 (51.3%) deaths within 30 days of the incidence of candidemia.

To assess whether the management of candidemia cases followed the Brazilian guideline we used only 80 of the patients who received treatment for analysis, and the outcome of mortality was defined as deaths occurring within 30 days.

Initial treatment with echinocandin was performed in 48 (60%) of the patients. Of those, 15 (31.2%) had death as the outcome (*p* = 0.39). Step-down to fluconazole was seen in twenty-one (26.2%) of the patients, with two cases (9.5%) eventuating in death (*p* = 0.03).

The collection of the first control blood culture occurred in 73 (91.2%) of the patients, but only in 43 (58.9%) did this collection occur within the first five days of candidemia. The rest of the collections occurred after the 5th day. Among patients with a first positive control blood culture (N = 19), fourteen (73.6%) underwent a second collection, with five (35.7%) being positive. Among those five, three (60%) had a third control blood culture; all were negative.

The Brazilian guideline recommends that the duration of treatment should be 14 days after the first negative blood culture, as long as there are no complications, and the patient is clinically stable; this was seen in 65.4% of cases. Some patients died before the fourteen days of treatment were completed. Of those patients who lived, 65.4% were treated 14 days after the first negative blood culture.

Removal of the central venous catheter occurred in 66 (82.5%) of the patients. However, analyzing the nineteen patients whose first control blood cultures were positive for *Candida* spp., 17 of them had their CVC removed posteriorly.

Table 2 summarizes the adherence to the Brazilian guideline for the treatment of candidemia.

Among 80 patients who received treatment, 28 cases (35%) evolved to death within 30 days.

We performed a univariate analysis correlating adherence to the guide and the 30-day mortality. Table 3 summarizes these results.

Central venous catheter removal and initial echinocandin therapy were more prevalent in the surviving group, but with no statistically significant difference. On the other hand, step-down to fluconazole demonstrated a better survival in the multivariate analysis: OR 0.15 (95% CI 0.03–0.8); *p* = 0.02.

## 4. Discussion

This study aims to assess the adherence to the current guidelines of the Brazilian Society of Infectious Diseases.

When we analyzed the findings, we observed that only 69.5% of patients received treatment. Even considering that 26.9% of patients did not have the chance to receive antifungal treatment because their cases eventuated to death within 3 days of the incidence of candidemia, we still had 3.5% of patients who did not receive any antifungal treatment. It is worth mentioning that positive blood cultures for *Candida* species are rarely interpreted as contamination, and all episodes of candidemia should be treated, considering the high morbidity and mortality of this infection [9]. We also did not evaluate the patients who died before the initiation of treatment, and since we do not have a severity score, we have no way of knowing if these patients were more critically ill and if they could have benefited from empirically based antifungal strategies.

Regarding the initial therapeutic regimen, we observed that in only 60% of the cases was echinocandin the drug of choice. This finding underscores the need for changing prescription habits and continuing education for physicians. Despite the fact that our univariate analysis did not show a positive correlation with survival (*p* = 0.392), there is clear evidence in the literature that initial echinocandin therapy reduces mortality, and for this reason it is the drug class of choice [10]. Also, considering the epidemiological change in the incidence of candidemia caused by *C. glabrata*, we emphasize the need for medical education as to the adequacy of the initial treatment with an echinocandin. In addition to our sample having 20% of its candidemia cases being caused by *C. glabrata*, we also have 46.1% of patients with the presence of gastrointestinal diseases, which may contraindicate the use of azoles [11].

On the other hand, the duration-of-treatment criterion had a good adherence level (65.4%), but this can be improved. Our findings show that treatment for 14 days after the first negative blood culture did not improve outcome (*p* = 0.186); however, this analysis was harmed by the fact that some patients died before having the chance to complete the treatment.

In our study, step-down to fluconazole was performed in 26.2% of the cases. Our guideline also recommends a step-down to oral azole therapy within 5–7 days as long as the patient has achieved a clinical stability characterized by resolution of signs and symptoms associated with the infection as well as clearance of the *Candida* (which should be susceptible to the azole) from the bloodstream [5,7]. The study by Husni supports step-down to oral azoles in patients with clinical and microbiologic resolution since patients who had a step-down presented a favorable outcome, even a more favorable outcome and better survival, compared to those patients who did not undergo any stepdown [*p* = 0.022] [12]. In our study we show that the step-down to fluconazole was safe (*p* = 0.004). Again, as we do not have a severity score, we cannot analyze whether the patients who underwent a step-down to fluconazole were less critical and because of that had better survival.

Removal of the CVC was the strategy with the highest adherence (82.5%), and this was even greater (89.4%) where there was a persistence of candidemia, demonstrating a good knowledge by the medical teams of the probable pathogenesis of candidemia and the management of these infections.

The CVC withdrawal strategy aiming to improve survival is still quite controversial in the literature. Several studies that analyzed risk factors for mortality demonstrate that CVC removal is beneficial, but Nucci et al. demonstrated that early CVC removal (within 72 h of candidemia) had no impact on six different outcomes, including mortality [13]. The Brazilian guideline recommends CVC replacement after 72 h, if positive hemoculture persists, with a C-III level of evidence and based on the article by Nucci et al. [13]. Due to the specification of this duration, we evaluated the percentage of CVCs removed within 72 h; this was performed in 77.2% of cases. Another question considers the source of the candidemia. If the source of candidemia is the central venous catheter, its removal is essential [14]. In our study, we did not prove a positive association between survival and CVC removal (*p* = 0.500), not even where the first blood culture was positive (*p* = 0.588). Although the sample size is very small, in our study, we had positivity of the CVC tip with concordance of the species isolated in the peripheral blood in 41.3% of the cases, demonstrating that the CVC was a possible source of the candidemia.

The control blood culture collection strategy also had good adherence, as we found 58.9% adherence to the first serial collection and 73.6% adherence where there was persistence of positive blood cultures, demonstrating that prescribers are well-oriented about this practice, but this also could be improved.

Regarding the performance of complementary exams, despite this being a controversial topic, we noticed a good level of adherence, especially regarding fundoscopy (73.7%). Earlier publications from the 1990s showed a prevalence of endophthalmitis ranging from 2.4 to 25.8%, which decreased to 0.9 to 7.9% in the following decade [15]. Considering that critically ill patients in ICUs will not report visual symptoms and given the poor prognosis of ocular sequelae, even with a very low incidence in our series (1.7%), we recommend performing this exam in line with the consensus [16]. Peçanha-Pietrobom and Colombo, in a review describing challenges in the clinical management of invasive candidiasis in critically ill patients, ask about the correct classification for ocular candidiasis, since one may have chorioretinitis, which is different from endophthalmitis [17]. The authors usually classify chorioretinitis as probable or possible invasive candidiasis while endophthalmitis (with vitreous involvement) is always considered to be proven or probable invasive candidiasis [18].

In our study, echocardiography was performed initially in 63.7% (TTE) and 11.2% (TEE) of cases, with an incidence of infective endocarditis (IE) found in 4.9% (4 patients). Separate analyses of these four cases showed that none of them had risk factors for endocarditis and two out of four had persistent positive blood cultures despite effective therapy. Therefore, we believe that the recommendation of the Brazilian consensus to perform echocardiography only if blood cultures are persistently positive is quite coherent [7]. Regarding which echocardiographic method to use, the ease of performing each method, the examiner’s experience and the question of whether or not the patient is able to undergo a more invasive examination should be taken into account. Considering the low prevalence of *Candida* endocarditis documented in most series, a more reasonable approach would be to replace universal screening with echocardiography by a targeted screening, only used for high-risk patients with clinical suspicion of endocarditis or if blood cultures are persistently positive [16].

The presence of infectious endocarditis was not associated with mortality (*p* = 0.521), but we had few cases of IE, which may contribute to this lack of association. As we only had one case of endophthalmitis, we could not perform an analysis.

Despite the small size of our sample and the fact that it is a single-center study, this is the only Brazilian study to analyze guideline adherence and outcomes of patients with candidemia. Another limitation of this study was that we did not perform susceptibility tests for the strains, due to their non-availability, especially that of *C. parapsilosis*, which has gained increasing importance due to azole resistance [19].

Several published studies have shown that both adherence to the guidelines and consultation with an infectious disease specialist improve the survival of patients with candidemia [20,21,22,23,24,25,26]. Table 4 summarizes the percentages of adherence to these studies.

Cuervo et al. [20] published a study in which they analyzed the impact of using the guidelines on mortality from the Infectious Diseases Society of America (IDSA) and the European Society of Clinical Microbiology and Infectious Diseases (ESCMID). They found that 81.7% showed compliance with more than 50% of the recommendations. A consistency rate of lower than 50% as to the recommendations was an independent risk factor for mortality.

Lehmann et al. [21] published a study analyzing adherence to IDSA guidelines in 157 cancer patients with candidemia, as mentioned above. This study had high rates of adherence to the guideline, but it is worth mentioning that in this study 95.5% of the cases had an Infectious Diseases consultation. The noncompliance was associated with poor cancer prognosis (*p* < 0.1). Despite the specific population, this study shows us that a better management of candidemia results in better patient prognosis.

Moni et al. [22] analyzed candidemia mortality before and after an implementation of a bundle of recommendations for a sample in India. However, the authors found an increase in adherence to the practices after the implementation of the bundle, although appropriate therapy, duration of treatment, and complementary exams were not associated with survival. Also, they failed to prove an association between CVC removal and mortality in a lower-middle income country setting.

Based on the ESCMID and IDSA guidelines, the European Confederation of Medical Mycology (ECMM) designed a score for clinical candidemia management named the EQUAL score. This score quantifies candidemia guideline adherence as a marker of treatment quality. The maximum score is 20 points for patients with CVC and 17 points for patients without CVC. It is a tool for facilitating the evaluation of guideline adherence [23].

Kim et al. [24] analyzed the impact of adherence to the guideline, defined by the EQUAL score, in South Korea. They found that the EQUAL score was lower in non-survivors’ groups, with a higher proportion of patients with a score < 15 (52.9% vs. 37%, *p* = 0.017) and use of CVC. Additionally, the mortality rates within 30 days showed a great difference between patients with an EQUAL score ≥ 15 (60.5% of survival) and patients with an EQUAL score < 15 (44.4% of survival), *p* = 0.003.

The same ECMM analyzed episodes of candidemia across Europe, and the EQUAL scores were used to assessed adherence to recommendations of the ESCMID and IDSA [25]. Different from our findings, here they found that initial treatment with echinocandin was associated with lower mortality. The EQUAL Candida scores were higher in patients who survived versus those who died.

There are several scores used to quantify compliance with the guidelines in candidemia management but evidence supporting the association of these scores with the prognoses is scarce.

In a study conducted by Calderon-Parra et al. [26], the authors evaluated different scores, and they concluded that the EQUAL score was associated with a decreased 30-day mortality.

On the other hand, adherence to the guidelines and their measurements is an important tool for the practice of antifungal stewardship. Antifungal stewardship is a coordinated approach to monitoring and directing the appropriate use of antifungal agents in order to achieve optimal clinical outcomes and minimize selectivity and adverse events. Prescriptions of antifungals may be optimized by considering the spectrum of action, pharmacokinetics and pharmacodynamics, duration of use and route of use, while also taking into account their high cost, the potential for toxicity and the need for expertise to direct clinicians in prescribing them [27].

A German study evaluated the impact of bedside antifungal stewardship on the clinical management and prognoses of patients with candidemia [28]. The adherence to recommendations of the ESCMID and IDSA guidelines improved after the intervention of the antifungal stewardship team. Antifungal treatment improved from 86% to 100%, and follow-up blood cultures improved from 40% to 71%. Overall mortality did not differ significantly; but the compliance of specific recommendations improved patient care and resulted in better candidemia management. So adherence to the guidelines, including the switch to fluconazole, is an important tool for practicing antifungal stewardship.

Our study did not use a score to assess adherence to the guideline, which was a limitation, but the analysis of these nine recommendations demonstrates that it is necessary to improve adherence to specific recommendations and also disseminate the strategies of initial use of echinocandin as the drug of choice, as well as guidelines relating to length of treatment, follow-up, and complementary exams. Our study provides reassurance that the step-down to fluconazole is safe and that it should be recommended, if the preexisting conditions are present. Lastly, our findings also suggest an opportunity for antifungal stewardship.

## Figures and Tables

**Table 1 jof-10-00282-t001:** Demographics and clinical characteristics of 115 patients with candidemia.

Characteristics	N (%)
Male gender	68 (59.1%)
Age (mean and variation in years)	55 (4–86)
Comorbidities	
Gastrointestinal tract diseases	53 (46.1%)
Chronic kidney diseases	32 (27.8%)
Diabetes mellitus	30 (26.1%)
Chronic pulmonary diseases	22 (19.1%)
Organ transplantation	22 (19.1%)
Neoplastic diseases	13 (11.3%)
Chronic heart diseases	13 (11.3%)
Immunodeficiency	7 (6.1%)
Pancreatitis	7 (6.0%)
Burns	4 (3.5%)
Risk conditions	
Central venous catheter (CVC) in the last 15 days	109 (94.7%)
Previous use of antibiotics in the last 30 days	104 (90.4%)
ICU admission in the last 15 days	94 (81.7%)
Surgery in the last 30 days	60 (52.2%)
Gastrointestinal surgery in the last 30 days	29 (25.2%)
Mechanical ventilation in the last 15 days	57 (49.5%)
Hemodialysis in the last 15 days	57 (49.5%)
Corticosteroid use in the last 15 days	57 (49.5%)
Immunosuppressant use in the last 30 days	26 (22.6%)
Total parenteral nutrition in the last 15 days	24 (20.9%)
Neutropenia in the last 30 days	6 (5.2%)

ICU: Intensive Care Unit; CVC: Central venous catheter.

**Table 2 jof-10-00282-t002:** Adherence to Brazilian guideline recommendations (N = 80).

Recommendations	
N (%)	
Initial treatment with echinocandin	48 (60%)
Step-down to fluconazole	21 (26.2%)
Collection of first control blood culture within 5 days of the incident candidemia	43 (58.9%)
Collection of second control blood culture, if the first one had been positive	14 (73.6%)
Treatment for 14 days after the first negative blood culture	53 (65.4%)
CVC removal	66 (82.5%)
CVC removal if the first control blood culture had been positive	17 (89.4%)
Performing a transthoracic echocardiogram	51 (63.7%)
Performing an eye fundoscopy	59 (73.7%)

CVC: Central venous catheter.

**Table 3 jof-10-00282-t003:** Guide adherence and 30-day mortality.

	Death (N = 28)	Survival (N = 52)	*p*
Initial treatment with echinocandin	15 (53.5%)	33 (63.4%)	0.392
Step-down to fluconazole	2 (7.1%)	19 (36.5%)	0.004
Time between candidemia and initial treatment (days)	2.46	4.28	0.089
Treatment for 14 days after the first negative blood culture	17 (60.7%)	39 (75%)	0.186
CVC removal	22 (78.5%)	44 (84.6%)	0.500
CVC removal if the first control blood culture had been positive	5 (17.8%)	12 (23%)	0.588
Infectious endocarditis	2 (7.1%)	2 (3.8%)	0.521
Endophthalmitis	0	1 (1.9%)	NA

**Table 4 jof-10-00282-t004:** Comparative analysis of the percentages of adherence to the guidelines published in different studies.

Reference	Number of Patients	Initial Therapy with Echinocandins	Follow-Up with Blood Cultures	CVC Removal	Duration of Therapy in Accordance with Guidelines	Fundoscopy	Stepdown to Fluconazole
Cuervo et al. [20]	445	62%	72%	79%	-	48%	-
Lehmann et al. [21]	157 cancer patients	99.2%	100%	90.1%	93.2%	78.8%	73.4%
Moni et al. [22]	175	87%	67%	89%	78%	-	-
Our study	80	60%	58.9%	82.5%	65.4%	73.7%	26.2%

## Data Availability

The data are contained within the article.

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
