# Peer review of "Guideline Adherence and Outcomes of Patients with Candidemia in Brazil"

_jof, 2024, doi:10.3390/jof10040282_

Round 1

Reviewer 1 Report

- Please mention how often the CVC is changed in this patients. Is this the effect of not changing the CVC on time?

-Authors should analyze treatment descalation or escalation by type of Candida specie.

-What do the Brazilian Society of Infectious Diseases guidelines say about number of days for CVC change? 

Please review spelling errors . I corrected a few of them 

Reviewer 2 Report

Dear Authors,

I read your manuscript concerning the guideline Adherence and Outcomes of Patients with Candidemia in Brazil. The paper is submitted as an original article and reports data from clinical practice in one centre. The current opinion is that, despite the guidelines, you have a non-neglectable index of other clinical practices. However, from an epidemiological point of view, it could be exciting and helpful in improving patient management. The paper is an original article, but some points must be improved.

1.      It is unclear where the study was performed. Is it a multicenter or single-center study? Add in the abstract, not just in the main text.

2.      Lines 19-21, the percentages reported are not clear.

3.      Lines 331-335 missing data, check

4.      Have you had relapse cases?

5.      Standard deviation is missing in tables

6.      Interestingly, gastrointestinal surgery in the last 30 days seems not to be related with candidemia while ah high percentage of patients with GI tract diseases had candidemia. Explain a possible rationale about this point.

7.      Lines 113 report all the isolated species.

8.      Lines 114 115 unclear ;

9.      Improve the antifungal stewardship section and the rational to switch at the fluconazole treatment.

10.  Lines 149-152, what Candida sp. have been isolated?

11.  Table 2. Add an abbreviation in the caption.

12.  IDSA acronym

13.  The references should be reported after the Author et al., not at the end of the sentence.

14.  The discussion and the results are too heavy for the readers. Add a table summarizing your data compared to the others (you cited a paper about guidelines adherence; a comparison could be optimal).

Reviewer 3 Report

The study of Machado Araujo et al. assessed adherence of clinicians to the Brazilian guideline for the management of candidemia in relation to nine outcomes. This is an interesting work, however, some major and minor issues should be addressed by the authors.

Major comments:

The first and most important issue that needs to be addressed is the time of the study. According to what is reported by the authors, the Brazilian guidelines were published in 2013, which is three years earlier than the IDSA guidelines on the management of candidiasis published in 2016. First, how did the Brazilian guidelines differ from the IDSA ones? Secondly, despite being a strong reference, guidelines are not the only document driving clinical decisions and therapy, as according to multiple epidemiological reports on candidemia have depicted an evolution of the different clinical scenarios from 2013 up to 2023, with a dramatic increase in non-albicans species and associated antifungal resistance, especially with C. parapsilosis and C. glabrata. These aspects should be addressed, especially trends in antifungal resistance. Plus the time span of the study period is another issue. Despite being an interesting work readers might wonder why was only the 2016-2018 period chosen as study period. Can this relatively short and not up to date period be representative of the actual adherence? Why wasn’t clinical practice evaluated across multiple years, considering that now we are in 2024. It would be therefore useful to evaluate evolution of guidelines adherence along with species distribution and antifungal resistance profiles across a longer time period.

The study of Machado Araujo et al. assessed adherence of clinicians to the Brazilian guideline for the management of candidemia in relation to nine outcomes. This is an interesting work, however, some major and minor issues should be addressed by the authors.

Major comments:

The first and most important issue that needs to be addressed is the time of the study. According to what is reported by the authors, the Brazilian guidelines were published in 2013, which is three years earlier than the IDSA guidelines on the management of candidiasis published in 2016. First, how did the Brazilian guidelines differ from the IDSA ones? Secondly, despite being a strong reference, guidelines are not the only document driving clinical decisions and therapy, as according to multiple epidemiological reports on candidemia have depicted an evolution of the different clinical scenarios from 2013 up to 2023, with a dramatic increase in non-albicans species and associated antifungal resistance, especially with C. parapsilosis and C. glabrata. These aspects should be addressed, especially trends in antifungal resistance. Plus the time span of the study period is another issue. Despite being an interesting work readers might wonder why was only the 2016-2018 period chosen as study period. Can this relatively short and not up to date period be representative of the actual adherence? Why wasn’t clinical practice evaluated across multiple years, considering that now we are in 2024. It would be therefore useful to evaluate evolution of guidelines adherence along with species distribution and antifungal resistance profiles across a longer time period.

Minor comments:

Introduction.

Line 33-38. It is important to present data from other studies regarding the current shifts in the candidemia epidemiology especially with non-albicans species as C. parapsilosis and C. glabrata and their associated rise in antifungal resistance.

Line 46-53. Please add a reference to the IDSA guidelines on the management of candidiasis and compare them with the Brazilian ones.

Line 68 “Candida sp” must be corrected with “Candida spp “ throughout the entire text.

Methods.

Line 87. “upon documentation of a favorable clinical response to treatment with echinocandis (i.e. negative blood culture);” which are the other factors included in the evaluation of treatment outcomes? Please add them in this section.

Results.

Why are data regarding antifungal susceptibility profiles missing? It would be appropriate if authors could briefly discuss results from antifungal susceptibility testing performed and present in a supplementary table.

Line 104. How many patients presented a concomitant bacterial bloodstream infection?

Line 113. please report other species involved.

Line 124-126. Revise gramma mistake (escalonated, were guide etc.)

Round 2

Reviewer 2 Report

Dear Authors, 

Most of the previous points have been addressed but not all. 

1) Interestingly, gastrointestinal surgery in the last 30 days seems not to be related with candidemia while ah high percentage of patients with GI tract diseases had candidemia. Explain a possible rationale about this point. As previously reported, you should report this point in the discussion because is not just about risk factor, but it is a key element in the therapeutic choise and the relative adherence (Spernovasilis N, Kofteridis DP. Pre-Existing Liver Disease and Toxicity of Antifungals. J Fungi (Basel). 2018 Dec 10;4(4):133. doi: 10.3390/jof4040133. PMID: 30544724; PMCID: PMC6309049.)

2) The references should be reported after the Author et al., not at the end of the sentence. Point not addressed. For example, Lehmann et al. [25]. 

3) Author Contributions, Institutional Review Board Statement, Informed Consent Statement, Data Availability Statement, Conflicts of Interest are mssing. Add. 

4) The discussion and the results are too heavy for the readers. Add a table summarizing your data compared to the others (you cited a paper about guidelines adherence; a comparison could be optimal. The Point was not addressed. In the comment for the reviewer you stated "they have different approaches and outcomes, we thought that a table would not be productive". So, what is the rationale for cite them in the discussion? A table is used to have the most important data compared to yours and to make and be more useful in clinical practice.

Reviewer 3 Report

Authors have addressed all major issues raised in the previous report and added a disclosure in the discussion section highlighting study limitations in regards of previous comments about issues that could not be modified. 

Authors have addressed all major issues raised in the previous report and added a disclosure in the discussion section highlighting study limitations in regards of previous comments about issues that could not be modified. The further explanations provided by the authors, especially those regarding study limitations are useful to better interpret the data provided. Despite some requested updates could not be implemented, the explanations given by the author along with the intrinsic valure of their reults make their study a usfeful resource tool in the context of fungal infectious diseases clinical practice.   

Round 3

Reviewer 2 Report

Dear Authors,

All the corrections have been made, and the paper has been improved. 

All the points have been addressed.